# Heparin-Induced Thrombocytopenia: A Review of New Concepts in Pathogenesis, Diagnosis, and Management

**DOI:** 10.3390/jcm10040683

**Published:** 2021-02-10

**Authors:** Matteo Marchetti, Maxime G. Zermatten, Debora Bertaggia Calderara, Alessandro Aliotta, Lorenzo Alberio

**Affiliations:** 1Division of Hematology and Central Hematology Laboratory, Lausanne University Hospital (CHUV) and University of Lausanne (UNIL), CH-1011 Lausanne, Switzerland; matteo.marchetti@ghol.ch (M.M.); Maxime.Zermatten@chuv.ch (M.G.Z.); Debora.Bertaggia-Calderara@chuv.ch (D.B.C.); Alessandro.Aliotta@chuv.ch (A.A.); 2Service de Médecine Interne, Hôpital de Nyon, CH-1260 Nyon, Switzerland

**Keywords:** heparin-induced thrombocytopenia (HIT), new concepts, pathogenesis, diagnosis, management, immune PF4/heparin/antibody complexes, Bayesian diagnostic thinking, therapeutic plasma exchange, intravenous immunoglobulins (IVIG)

## Abstract

Knowledge on heparin-induced thrombocytopenia keeps increasing. Recent progress on diagnosis and management as well as several discoveries concerning its pathogenesis have been made. However, many aspects of heparin-induced thrombocytopenia remain partly unknown, and exact application of these new insights still need to be addressed. This article reviews the main new concepts in pathogenesis, diagnosis, and management of heparin-induced thrombocytopenia.

## 1. Introduction

Heparin-induced thrombocytopenia (HIT) is a fascinating, complex, and still partially obscure immunological syndrome. It is associated with a very high prothrombotic risk and may cause limb and life-threatening complications. Thus, a rapid accurate clinical and laboratory recognition as well as a prompt and effective management are required. Since its discovery, knowledge on HIT has been impressively growing, and all fields have been evolving: from pathogenesis to diagnostic approaches and management. In this review, we will particularly focus on the most relevant clinical new insights and advances on HIT.

## 2. Pathophysiology

The pathophysiology of heparin-induced thrombocytopenia (HIT) is complex and far exceeds simple platelet activation. Its spectrum is broad. Here, we will describe each step of HIT pathophysiology, beginning with antigen formation, followed by immune reaction and antibody synthesis, development of a severe prothrombotic state, and ending with thromboembolism, which has the potential to be life threatening.

### 2.1. Antigen Formation: Platelet Factor 4-Heparin (PF4/H) Complexes

The HIT antigen is situated on the platelet factor 4 (PF4), a chemokine that is contained in platelet α-granules. PF4 is not immunogenic in its primary form. Conformational PF4 changes are needed to expose a neo-epitope, which is the HIT antigen. These changes occur by the formation of complexes between PF4 and negatively charged molecules, especially heparin and other glycosaminoglycans (GAGs) [1]. The size and the charge of the complexes play a central role in pathogenicity. These two parameters depend on the relative amounts of PF4 and heparin. The PF4/heparin complexes are electrostatically formed by at least 16 PF4 molecules (positively charged) assembled with heparin chains (negatively charged) in multimolecular ultralarge complexes (ULCs), which will participate to platelet activation (see below) [2]. Quantitatively, the maximal amount of these ULCs is formed at equimolar PF4/heparin ratio [2,3]. Of note, when performing these analyses in presence of platelets, the ratio has been reported to be 20:1, indicating that, in vivo, a proportion of the heparin chains could be replaced by endogenous glycosaminoglycans in the glycocalyx of the cell surface [4]. Importantly and compared to unfractionated heparin (UFH), far fewer ULCs are formed with low molecular weight heparins (LMWH), and none with fondaparinux [2], because heparin chains with at least 12 saccharides are necessary to form ULCs [2,5]. This explains the lower risk to develop HIT with LMWH. Qualitatively, the immunogenicity of the complexes depends on their net charge and not their size. A positive net charge facilitates the interaction with immune cells [3]. Therefore, the highest immunogenicity is reached when PF4—in absence of platelets (see above)—is in excess and at a molar PF4/heparin ratio of 20/1 [3]. Of note, these mechanisms explain why fondaparinux is immunogenic but only rarely has been reported as a cause HIT [6]. Similarly, the immunogenicity of a high PF4/heparin ratio also explains the high incidence of HIT in patients with high amounts of circulating PF4 (e.g., orthopedic and vascular surgery) and prophylactic doses of UFH.

### 2.2. Anti-PF4/H Antibodies Synthesis

Innate and adaptive humoral and cellular reactions lead to anti-PF4/H antibodies synthesis. Briefly, once formed, ULCs activate complement, leading to the deposition of C3/C4 on the complexes. This allows their binding to cluster of differentiation 21 (CD21, the complement receptor 2) on B cells, which facilitates their activation, antigen transport to secondary lymphoid follicles, and antigen transfer. This culminates in an adaptive humoral immune response and anti-PF4/H antibodies production [7].

### 2.3. Antigen Formation on the Platelet Surface and Platelet Activation

The antigenic complex formation occurs on the platelet surface in a dynamic and potentially reversible manner. In presence of PF4, increasing heparin initially leads to an increasing antigen-complex size. HIT antibodies will then bind to these ULCs [8]. With further increase, heparin would then displace PF4 from the platelet surface and diminish the size of the antigen-complexes, thus decreasing their capacity to activate platelets [8]. The antigen–antibody binding on the platelet surface induces platelet activation via FcγRIIa (CD32, the low affinity IgG receptor) and leads to platelet degranulation and aggregation. Degranulation increases the available PF4 concentration for further antigen-complex formation [9,10]. Besides these “classical” platelet activation endpoints, platelet activation also induces the production of procoagulant platelets and platelet-derived procoagulant microparticles [11], dramatically enhancing thrombin generation. Moreover, in presence of an excess of PF4, further cells, including monocytes, endothelial cells, and neutrophils, can be recruited. The interactions and roles of these cells are summed up below and graphically presented in Figure 1.

### 2.4. Other Cells Beyond Platelets Involved in HIT Prothrombotic State

#### 2.4.1. Monocytes

Monocytes participate in HIT hypercoagulability. Indeed, PF4 binds monocytes with a higher affinity than platelets. This is due to the different GAGs proportion in their glycocalyx and to a variable PF4 affinity for the different GAGs. Indeed, PF4 binds with decreasing intensity to heparin, heparan sulfate, dermatan sulfate, chondroitin-6-sulfate, and chondroitin-4-sulfate [12]. This facilitates the binding of HIT antibodies [13,14,15,16], which appears to be optimal in absence of heparin [13]. This could be due to the higher availability of PF4, which is not bound by heparin. The formation of antibody/antigen complexes on the monocyte surface leads to their activation [15], inducing (i) secretion of IL-8 [13] and surface expression of tissue factor (TF) [13,16] and (ii production of tissue-factor expressing microparticles (TF-MPs) [13]. TF expression seems to depend on FcγRIIa [16], while the de novo synthesis and production of TF-MPs appears to depend on FcγRI [13]. Moreover, monocyte activation leads to their glycocalyx sulfation, which induces a higher affinity for PF4 binding, thus sustaining an amplification loop [14]. These changes provoke a procoagulant activity of the monocytes [17], culminating in thrombin generation. The generated thrombin will in turn activate platelets in other amplification loops leading to procoagulant platelets due to platelet co-activation via FcγRIIa [18], whose downstream signaling is similar to that observed upon glycoprotein (GP) VI engagement [19]. Of note, monocyte activation can also occur via P-selectin expressed on platelets [20]. To summarize, monocytes are activated by antigen/antibody HIT complexes, which lead to thrombin generation via TF expression and TF-MPs production.

#### 2.4.2. Endothelial Cells

HIT antibodies bind to PF4 on endothelial cells as well [21,22], possibly via the F(ab) region of the antibodies [22]. This binding, which seems to be preferentially directed towards microvascular endothelial cells, requires an endothelial preactivation, possibly caused by TNFα released during platelet activation [22]. Because PF4 is required, the primary step remains platelet activation and degranulation. This leads to the required high PF4 concentrations that exceed the PF4-neutralization capacity of heparin. In presence of vascular lesions, PF4 could mainly bind to endothelial cells [23], and these lesions could be the preferred sites for thrombosis development in HIT [24]. The formation of antigen–antibody complexes on the cell surface induces additional endothelial injuries and activation [21,22,25], which leads to the expression of TF [21] and adhesion molecules [22] and to changes in the glycocalyx of the endothelial cells. Indeed, injured endothelial cells seem to release thrombomodulin [22,25], which could lead to locally decreased anticoagulant potential. Of note, the binding of PF4 could be more intense on injured endothelial cells [23], despite loss of negative charges. This could indicate the existence of a high-affinity PF4-binding site being unveiled and inducing a positive feedback loop with increased formation of immune complexes and sensitization of neighbor endothelial cells [23].

Moreover, PF4 could bind to extended strings of von Willebrand factor, which have been released from activated or injured endothelial cells, consecutively exposing the HIT antigen on bound PF molecules [26]. This is recognized by HIT antibodies leading to the formation of a IgG Fc-rich network with complement activation and activation of additional cells.

#### 2.4.3. Neutrophils

Neutrophils are thought to play an essential role in HIT hypercoagulability and in thrombosis development. They are activated by P-selectin on platelets and via FcγRIIa [27] by anti-PF4/H antibodies and heparin immune complexes formed on their surface, which are bound to chondroitin sulfate [28]. This induces neutrophil extracellular traps (NET) formation and release (NETosis). NETs participate in the hypercoagulability in HIT. First, they can bind to PF4, developing a surface rich in fragment crystallizable regions (Fc-domain) and an amplification loop [29]. Second, they are thought to be essential for thrombosis development [27], via their procoagulant activities, in particular direct activation of the contact phase of thrombin generation.

### 2.5. HIT, A Side Effect of an Immune Mechanism?

PF4 and anti-PF4/polyanion antibodies could play a physiologic role in the host immune defense, particularly against Gram-negative bacterial infections. Indeed, bacteria can activate platelets, leading to degranulation and release of PF4 and polyphosphates. PF4 and polyphosphates can successively bind on lipid A on Gram-negative bacteria [30,31,32], leading to opsonization of bacteria by anti-PF4/polyanion antibodies, enabling better host defense against infection. The formation of anti-PF4/heparin antibodies would therefore be a pathological molecular-mimicry activation of a physiologic mechanism.

### 2.6. Importance of Specific Gene Polymorphisms

As widely known, not all patients developing HIT antibodies develop HIT, and not all patients with HIT develop thromboembolic complications. The mechanisms underlying these variable responses are not completely understood. However, specific gene polymorphisms could be involved, especially in the risk to develop thromboembolic complications in HIT. Indeed, an association between the polymorphism of FcγRIIIA 158VV [30] and occurrence of HIT and between the polymorphism of FcγRIIA 131R and thromboembolic complications in HIT [31,32,33] has been observed in different studies. Further genetic polymorphisms could be identified in the future, explaining the different responses to HIT antibodies.

### 2.7. HIT without Heparin

Heparin is the most frequent negatively charged molecule that induces HIT. However, some patients develop symptoms and signs of HIT without exposition to heparin, especially after orthopedic surgery, which is known as spontaneous or autoimmune HIT (aHIT) [34,35,36,37,38]. This could be due to the presence of others polyanions that can induce “HIT”. This is the case for chondroitin sulfate [39,40], polyphosphates [41], nucleic acid [42], or bacterial components [43]. A brief comparison between classical HIT and aHIT is presented in Table 1. For further details on aHIT, we refer the reader to the excellent review of Greinacher, Selleng, and Warkentin [44].

To summarize, HIT is a complex immune-mediated pathology. Its mechanisms depend on the concentrations of PF4 and heparin, particularly their ratio to each other and involve platelets, monocytes, endothelial cells, and neutrophils as well. The activation of these cells induces, besides thrombocytopenia, a coagulation cascade activation leading to a severe hypercoagulant state.

## 3. Diagnostic Approach

Making a correct and rapid diagnosis of HIT is challenging and of utmost importance [45]. It requires the association of clinical parameters to estimate the pre-test probability and laboratory assays to confirm or infirm the diagnosis [46].

### 3.1. Clinical Pre-Test Probability

At the bedside, the cornerstones are thinking of HIT when appropriate (e.g., fall in platelet count, thrombosis despite heparin anticoagulation) and subsequently assess the clinical pre-test probability of HIT with validated clinical scores. To do so, the 4T score (thrombocytopenia, timing, thrombosis, other causes of thrombocytopenia; 0–8 points) has been developed [47], and its use is currently recommended by the American Society of Hematology (ASH) [48]. The HEP score is a more recent clinical score [49]. Compared to the 4T score, it showed a higher specificity among ICU patients [50]. However, it still needs broader implementation studies and is not yet recommended [48,50]. The clinical probability can rule out HIT or establish the indication for laboratory testing. Currently, it is considered that a low (0–3 points) 4T score can rule out HIT, while an intermediate (4–5) or high (6–8) score requires laboratory testing [48]. However, different studies observed HIT cases despite low 4T scores, which raised concern about ruling out HIT among patients with a 4T score of 3 [51,52,53,54].

### 3.2. Laboratory Work-Up

Different types of laboratory assays exist. In this review, we will focus on two platelet-activation assays (i.e., the serotonin-release assay (SRA) and the heparin-induced platelet aggregation test (HIPA)), on some broadly available immunoassays (IA) for HIT, and on emerging diagnostic strategies.

#### 3.2.1. Platelet-Activation Assays: The Gold Standard for HIT

Two platelet-activation assays using donor washed platelets are considered as diagnostic gold standards for HIT [55]. These assays are considered to be functional because they detect heparin-dependent platelet activation by either aggregation (HIPA) or degranulation and release of serotonin (SRA) when the patient’s plasma, possibly containing anti-PF4/heparin platelet-activating antibodies, is added in presence of pharmacological concentrations of heparin. However, these assays are technically demanding, time consuming, and unavailable for many hospitals, which delays definitive diagnostic work-up and optimal management [52]. As a consequence, their use is currently only recommended after a first clinical-biological work-up, namely among patients with intermediate and/or high clinical suspicion (i.e., 4T score > 3 points) and a positive IA [48]. Moreover, recent studies described the possibility of false-negative functional assays [52,56,57]. Ticagrelor can induce false negative HIPA among HIT patients, and other anti-platelet drugs might also cause false negative HIT functional assays [56]. Another mechanism is the novel concept of seroconversion preceding functional positivity in vitro [58]. To address this lack of sensitivity, several methods to avoid false-negative functional assays have been reported, but not validated by a dedicated study: (i) removing/inhibiting antiplatelet agents in the first case and (ii) performing a PF4-enhanced assay (PF4-SRA) [59,60] when suspecting the second case, although this approach could lead to false positive results [56,57,58,61]. Although such methods increase the performances of the gold standards functional assays, HIPA and SRA, false negative-negative results are still possible. Therefore, when the 4T score is high and the IA are strongly positive, HIT should still be considered despite a negative functional assay [45,48]. Apart from the SRA and the HIPA assays, many other platelet-activating assays have been developed, and for a more detailed review on this topic, we refer to the recent review of Tardy et al. [62].

#### 3.2.2. Immunoassays (IA): Rapid and Broadly Available Alternatives

Many laboratory techniques detecting anti-PF4/heparin antibodies exist.

##### The Classical IA

Enzyme-linked immunosorbent assays (ELISA) were the first wide available techniques [46]. They are nowadays still broadly used, and the ASH guidelines recommend their use as first-line laboratory test [48]. They are recognized to be highly sensitive (i.e., allow to accurately rule out HIT), but unspecific (i.e., leading to false-positive results and unnecessary non-heparin anticoagulation while awaiting the definitive result of a functional assay or for long course) [46,58]. Recently, Warkentin et al. highlighted that ELISA are not considered as rapid immunoassays anymore [58]. In addition, they underscored the original observation of Lindhoff-Last [58], that IgG-specific ELISA are more specific without being less sensitive for HIT, echoing a communication of the scientific and standardization committee of the International Society on Thrombosis and Haemostasis (ISTH) [63] in favor of use of IgG-specific immunoassays [58]. In a review published in 2017, Arepally also underscored that anti-PF4/H IgG antibodies are the most relevant isotype for HIT pathogenesis, IgA and IgM contribution to HIT being subordinate [46].

##### Other “Rapid” IA

Many other techniques detecting anti-PF4/H antibodies have been developed and studied. In 2016, two very relevant meta-analyses about rapid IA for HIT were published. Nagler et al. identified five tests as highly sensitive and specific for HIT, namely the polyspecific ELISA with an intermediate threshold (Genetic Testing Institute, Asserachrom), particle-gel immunoassay (PaGIA), lateral flow immunoassay (LFIA), polyspecific chemiluminescent immunoassay (CLIA) with a high threshold, and IgG-specific CLIA with a low threshold [64]. Sun et al. reached the conclusion that PaGIA, IgG-specific CLIA, and LFIA showed excellent sensitivity and specificity [65].

##### Emerging Diagnostic Strategies

Moreover, both meta-analyses highlighted that combining results of some of the different aforementioned rapid IA might improve diagnostic performance for HIT and thus further improve care in patients with suspected HIT. Additionally, Sun et al. highlighted that combining clinical assessment (i.e., pre-test probability) with rapid immunoassays in a Bayesian approach was likely to be the most powerful way to estimate an overall likelihood of HIT in real-world clinical practice [64,65]. In 2018, the ASH guidelines identified “integration of emerging rapid immuno-assays into diagnostic algorithms” as a pressing research priority [48].

Regarding the Bayesian diagnostic approach, Nellen et al. showed in 2012 for the first time ever that the combination of clinical pre-test probability (assessed by the 4T score) with the quantitative result of a rapid immunoassay detecting anti-PF4/heparin antibodies was valuable not only for excluding [66], but also for predicting a positive heparin-induced platelet aggregation test, i.e., for diagnosing HIT rapidly [51]. Other groups confirmed that a Bayesian diagnostic approach for HIT that combines clinical probability assessed with validated scores with rapid immunoassay results was a promising approach towards improvement of HIT diagnostic work-up, especially when semi-quantitative immunoassay results were used [53,54,67,68].

Recently, two groups described diagnostic approaches that combine two rapid immunoassays. Our group developed and prospectively validated a Bayesian diagnostic algorithm (the “Lausanne algorithm”) based on the 4T score and the combination of IgG-specific CLIA and PaGIA that correctly classifies >95% of patients within a 60 min laboratory work-up time [52]. This algorithm was recognized by Cuker and Cines to perform better than the ASH algorithm, correctly classifying “all patients with HIT and 95.4% of patients without HIT, whereas the ASH algorithm correctly classified only 91.1% of the patients with HIT and 93.2% of patients without HIT” [45]. Warkentin et al. described a diagnostic laboratory scoring system (0–6 points) based on the combination of IgG-specific CLIA and latex immunoturbidimetric assay (LIA) that reached a 99% sensitivity when both assays (CLIA and LIA negative threshold <1.0 U/mL) were negative and a positive predictive value for platelet-activating antibodies (i.e., positive SRA or PF4-SRA) of 97.1% for scores ≥5 points [69]. Of particular relevant note, these diagnostic approaches use the PaGIA and the IgG-specific CLIA or the LIA as rapid IAs. Thus, integrating other IAs in such Bayesian approaches that combine two rapid IAs still needs to be studied.

## 4. Management of Acute HIT

As described previously, HIT is a complex and severe prothrombotic state. Briefly, current cornerstones of HIT management are (i) immediate cessation of any heparin administration and (ii) introduction of non-heparin therapeutic anticoagulation, such as argatroban or bivalirudin (direct parenteral thrombin inhibitors), or danaparoid or fondaparinux (indirect parenteral factor Xa inhibitors) [48]. Recently, direct oral anticoagulants (DOACs) are emerging as an alternative in acute HIT or HIT with thrombosis, but the data on their use remain very limited [70]. For a detailed review and updated recommendation on the use and monitoring of these different non-heparin anticoagulants, we refer the interested readers to Table 2 and a recent comprehensive article [70]. In the present review, we will focus on other emerging and second-line management strategies in HIT that either directly target platelet-activating anti-PF4/heparin antibodies or inhibit antibody-mediated platelet activation (Table 3).

### 4.1. Therapeutic Plasmapheresis (TPE)

Experience with TPE in patients with HIT remains limited, as underscored in 2018 by Cho et al. who performed a national survey (USA) of academic apheresis services regarding practices in managing patients with HIT and found only 15.4% of respondents reporting having performed TPE on patients with HIT during the past year [83].

To date, case reports are the major source of evidence. Additionally, although a single TPE removes about two-thirds of the HIT antibodies [71], the exact number of required plasma exchanges in order to effectively lower the plasmatic HIT antibodies concentration and the duration of action of TPE due to mobilization of the extravascular antibody compartment remain unknown and difficult to predict. Thus, TPE is probably best guided by biological monitoring [72,73].

TPE has been described as an effective option for managing HIT in three particular situations [72]. First, pre/perioperative plasmapheresis along with intraoperative heparin anticoagulation is one of the three management options for patients with acute HIT or persisting positive functional assay and anti-PF4/H antibodies who require immediate cardiovascular surgery [48]. When doing so, sensitive functional platelet aggregation assays are best suited to determine readiness for heparin re-exposure [84]. The second situation is when non-heparin anticoagulation is contraindicated because of a major bleeding event, such as intracerebral hemorrhage (ICH) [79]. The third situation is when HIT clinical and biological course worsens/does not improve under well-conducted non-heparin anticoagulation (i.e., refractory HIT). For further details, we refer the reader to the recent excellent review of Onuoha et al. [72].

### 4.2. Intravenous Immunoglobulin (IVIG)

In the latest guidelines on HIT management, elucidation of the role of IVIG treatment in acute HIT has been defined as a key research priority for the management of HIT [48].

Regarding safety of IVIG use, a consensus of 15 Canadian hematologists stated in 2007 that IVIG were contraindicated for treatment of HIT because of a potential increased thrombotic risk [85]. However, in their recent study, Dhakal et al. found no increased risk for arterial or venous thrombosis incidence among patients with HIT treated with IVIG [86].

In clinical practice, many groups have reported favorable experience with this treatment, especially in patients with refractory HIT [72,87]. Recently, Warkentin underscored IVIG use as an adjunctive treatment, especially when thrombocytopenia persists in the setting of auto-immune HIT (aHIT) [74]. Briefly, aHIT is a subgroup of HIT including different disorders and is characterized by in vivo and in vitro heparin-dependent and heparin-independent platelet-activating antibodies. Because of high-titer, heparin-independent, platelet-activating antibodies, clinical course of aHIT is often severe and characterized by worsening or persisting HIT, even after beginning an alternative anticoagulation [74]. For a detailed clinical and biological description of aHIT, we refer to the excellent review of Greinacher, Selleng, and Warkentin [44].

Interestingly and in relationship with elements discussed above, in vitro IVIG have been shown to preferentially inhibit heparin-independent platelet-activating antibodies, explaining thus the rationale of their use in aHIT and/or refractory HIT [74]. Even if their exact mechanism of action remains unknown, they are believed to inhibit platelet (and other cells, see above) activation through FcγRIIa receptors [88].

### 4.3. New Insights of the Translational Research: The Quest for Inhibiting FcγRIIa-Mediated Platelet Activation

A chimeric anti-PF4/H antibody composed of a mouse IgG1 and a human Fc fragment was developed, namely the 5B9. It was obtained after immunization of transgenic mice with heparin and purified human PF4. 5B9 was proven to (i) induce platelet degranulation and release of serotonin when added in whole blood of healthy donors containing UFH (i.e., result in a positive SRA) and (ii) induce thrombocytopenia and thrombin generation when administered with UFH to another species of transgenic mice expressing human FcγRIIa and PF4 (i.e., cause HIT). Thus, 5B9 was described to fully mimic the effects of human HIT antibodies [89].

One emerging and appealing option that still needs further study is the IgG-degrading enzyme of *Streptococcus pyogenes* (IdeS), which cleaves a region that is critical in the interaction of IgG with FcγRIIa and thus disables platelet activation via this pathway [77]. Kizlik-Masson showed in vitro that IdeS selectively prevented platelet activation in the presence of heparin and 5B9 or human anti-PF4/heparin platelet-activating IgG antibodies without altering platelet aggregation induced by ADP and/or collagen. Moreover, the same group showed in vivo that IdeS prevented thrombocytopenia and thrombin activation (i.e., HIT) when 5B9 and UFH were administered to transgenic mice expressing human FcγRIIa and PF4 [77]. Summarizing, IdeS has already been studied in different mice models (immune thrombocytopenia, IgG-dependent glomerulonephritis, IgG-dependent arthritis, IgG-HIT models) showing promising effects [77]. Thus, in the field of HIT, IdeS is a promising emerging novel tool that might expand the available therapeutic arsenal.

## 5. Management of Patients Requiring Cardiovascular Surgery

On the one hand, according to Selleng et al. [90] and Warkentin et al. [91], anticoagulation with unfractionated heparin (UFH) for patients undergoing cardiac surgery in patients with a history of HIT is safe and effective, if circulating anti-PF4/heparin antibodies are no longer detectable [90,91,92]. On the other hand, management of patients requiring cardiovascular surgery and who have either (i) acute HIT, (ii) persisting positive functional assay and anti-PF4/heparin antibodies, or iii) negative functional assay, but persisting anti-PF4/heparin antibodies, is very challenging because the perioperative balance between both thrombotic and bleeding risk is fragile and can lead to fatal complications.

### 5.1. Patients with Acute HIT or Persisting Positive Functional Assay and Anti-PF4/Heparin Antibodies

To date, the ASH recommends delaying cardiovascular surgery among these patients. Because of a low level of evidence, there are only suggestions for management strategies in patients requiring immediate cardiovascular surgery [48].

The three main management options in these patients are (i) alternative intraoperative anticoagulation with bivalirudin, (ii) intraoperative anticoagulation with heparin and simultaneous antiaggregation with a potent platelet inhibitor (iloprost or tirofiban), or (iii) intraoperative anticoagulation with heparin and peri-operative plasma exchanges (see above) [48].

Bivalirudin is considered a safe alternative option for intraoperative anticoagulation in patients who undergo cardiovascular surgery when the interdisciplinary team is experienced and familiar with this technique, in particular avoiding blood stasis in the extracorporeal circuit and monitoring bivalirudin [93].

Alternatively, anticoagulation with intraoperative heparin and simultaneous short-acting and reversible anti-aggregation seems to be a valid strategy. In this context, apart from tirofiban (GP IIb/IIIa receptor blocker, half-life of 1.4 to 2.2 h, dependent on renal function) [94] and iloprost (synthetic analogue of epoprostenol PGI2 inhibiting platelet aggregation, adhesion and release reaction, half-life of 30 min) [95], cangrelor is a recent, potent, rapid-acting and reversible ADP receptor P2Y12 inhibitor with a very short half-life of 3–6 min [75]. Its use has been reported to be successful by different groups [81,96]. However, Scala et al. observed that cangrelor unreliably inhibits heparin-induced platelet aggregation in vitro when anti-PF4/heparin platelet-activating antibodies are present, concluding that cangrelor should not be used for HIT patients undergoing cardiac surgery unless its efficacy was confirmed in a particular patient with a presurgery negative aggregation test [82]. Cangrelor has a theoretical optimal profile (potent P2Y12 antagonist, rapid and very short-acting, reversible), but it needs to be further studied before being recommended or advised against.

A novel approach with limited published experience thus far is the use of IVIG to prevent HIT antibodies activating platelets and other cells as well (see above), possibly combined with additional cangrelor, in order to perform cardiovascular surgery with heparin [93].

### 5.2. Patients with Negative Functional Assay and Persisting Anti-PF4/Heparin Antibodies

To date, the ASH recommends with low level of evidence to favor intraoperative heparin for these patients on the basis of a few cases series [48,90,91]. However, SRA may not be sensitive enough to rule out the presence of pathogenic antibodies before cardiac surgery. Indeed, PF4-enhanced SRA has been described to be more sensitive than SRA, and since cardiac surgery induces a burst in PF4 plasma concentration, elevated intraoperative PF4 plasma concentrations might result in a positive SRA, mimicking thus in vivo a PF4-enhanced SRA [97]. HIT recurrence among patients with persisting anti-PF4/heparin antibodies who receive a single heparin dose is possible, especially if antibodies levels are high [98] and close medical and platelet follow-up is essential. Scala et al. reported the successful use of intraoperative heparin and simultaneous cangrelor in a patient with initial negative functional assay and persisting anti-PF4/heparin antibodies. They observed subsequent anti-PF4/heparin antibodies rise and positive functional assay seroconversion, concluding that even a short heparin administration (<90 min) induced boosting of clinically relevant anti-PF4/heparin antibodies. For similar patients, Scala et al. [76] and Warkentin et al. [91] underscored that (i) postoperative monitoring of platelet counts [76,91] and anti-PF4/heparin antibodies [76] is mandatory, even after single intraoperative heparin exposition [76,91], (ii) performing a functional assay is necessary when anti-PF4/heparin antibodies rise and/or platelet count decreases [76,91], and (iii) a platelet fall [76,91] or increase in D-dimers [76] must be considered as presumptive recurrent HIT and requires, thus, beginning of non-heparin anticoagulation [76,91]. It remains to be verified whether the administration of IVIG in this context may modulate HIT antibodies boosting [93].

## 6. Conclusions

In this review, we described novel insights in the pathophysiology of HIT, recent improvements for the accuracy and speed of its diagnostic work-up, as well as new concepts for its management.

The current model of HIT is that of a multi-step immune pathology in response to the simultaneous exposition to an endogenous (i.e., PF4) and an exogenous molecule (i.e., heparin), at specific molar ratios. Moreover, the importance of platelets has been well known for years, but the central roles of other cells (monocytes, endothelial cells, and neutrophils) and specific receptors have only been recognized a few years ago. The comprehension of the pathophysiology of HIT is key to improve the therapeutic arsenal and the management of patients. This is well shown by the opportunity offered by IVIG and IdeS and their inhibitory effects in the interaction of IgG and FcγRIIa. A further and more precise comprehension of the pathophysiology could allow to develop more specific therapy for HIT.

Clinically, the work-up of HIT has improved in the past few years with the emergence of rapid and accurate diagnostic algorithms combining clinical parameters (4T score) and rapid quantitative laboratory assays. It is now possible to rapidly and accurately rule in or out HIT. This drastically improves the management of patients with HIT suspicion.

In conclusion, our knowledge about HIT is rapidly improving, which leads to a better management of this pathology and a better prognosis for patients.

## Figures and Tables

**Figure 1 jcm-10-00683-f001:**
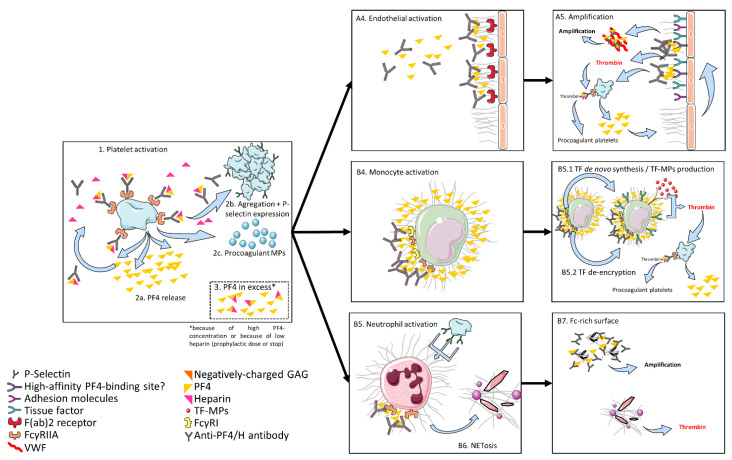
Pathophysiology of HIT. Platelets are activated by ultralarge complexes (ULCs) of PF4 and heparin via FcγRIIa. The activation leads to PF4 release, aggregation, P-selectin expression, and to production of procoagulant microparticles. After platelet activation, PF4 is in excess compared to heparin and can bind to endogenous glycosaminoglycans (GAG) on endothelial cells, monocytes, and neutrophils. Complexes of PF4 and GAG are recognized by HIT antibodies with consecutive activation of the cells. The activation of monocytes and endothelial cells leads to thrombin generation via expression of tissue factor (TF) and microparticles with TF (TF-MP), with further activation of platelets, creating a positive feedback loop. Moreover, the activation of endothelial cells leads to von Willebrand factor (VWF) secretion, on which PF4 binds, creating a Fc-rich surface, which leads to further activation of platelets. Activation of neutrophils leads to NETosis offering a Fc-rich surface with further activation of platelets and supporting thrombin generation (see text).

**Table 1 jcm-10-00683-t001:** Comparison of classical and autoimmune HIT.

	Classical HIT	Autoimmune HIT
**Antigen**	Neoepitope on PF4, revealed by a conformational change due to binding to a negatively-charged surface	
**Negatively-charged molecule**	Heparin	Other polyanions: chondroitin sulfate, polyphosphates, nucleic acid, bacterial components
**Pathogenesis**	Similar (see text: part 1a–d)	
**Therapy**	Heparin avoidance, alternative non-heparin anticoagulation	Intravenous immunoglobulins (IVIG), plasmapheresis

**Table 2 jcm-10-00683-t002:** Key information on alternative parenteral non-heparin anticoagulants (adapted from [70]).

	Argatroban	Bivalirudin	Danaparoid	Fondaparinux
Mechanism of action	Direct thrombin inhibitor	Direct thrombin inhibitor	Indirect factor Xa inhibitor (mediated by antithrombin)	Indirect factor Xa inhibitor (mediated by antithrombin)
Half-life	45 min	25 min	19–25 h	17–21 h
Route of administration	IV	IV	IV/SC	SC
Main elimination pathway	Hepatobiliary	Proteolytic (80%) and renal (20%)	Renal	Renal
Monitoring	Calibrated diluted anti-factor IIa assay	Calibrated diluted anti-factor IIa assay	Calibrated anti-factor Xa assay	Calibrated anti-factor Xa assay

Legend: IV, intravenous, SC, subcutaneous.

**Table 3 jcm-10-00683-t003:** Key information on second-line treatments of HIT.

	Therapeutic Plasmapheresis (TPE)	Intravenous Immunoglobulins (IVIG)	Cangrelor	IdeS (Imlifidase)
Rationale	Removal of pathological antibodies	Binding of pathological antibodies	Rapid-acting, reversible platelet inhibitor	Endopeptidase specifically cleaving IgG antibodies
Dose and mode/frequency of administration	A single TPE removes about 2/3 of the HIT-antibodies [71]. Exact number of TPE and duration of action unknown [72,73]	1 g/kg/day IV for 2 consecutive days. NB: use calculated dosing-weight among obese patients [74]	30 μg/kg IV bolus, followed by 4 μg/kg IV infusion [75,76]	0.24–0.5 mg/kg [77,78]. NB: Appropriate dosage has to be established in HIT
Indications	1. Acute HIT or positive functional assay + immediate cardiovascular surgery with heparin [48] 2. Alternative anticoagulation contraindicated [73,79] 3. Clinical course worsening despite non-heparin anticoagulation [72]	1. Clinical course worsening despite non-heparin anticoagulation [74,80] 2. aHIT [74,80] 3. Alternative anticoagulation contraindicated	Cardiovascular surgery when anti-PF4/heparin are present and intraoperative heparin use is mandatory [76,81,82]	To date, no clear indication
Pros	Effective for managing HIT in many conditions, regional anticoagulation possible	Favorable experience, especially in patients with refractory HIT and aHIT [74,80]	Favorable experience with this treatment	Promising results in mice HIT model [77] and kidney transplant patients [78]
Cons	Cost; infectious, metabolic complications; difficulty in predicting how often and how long to perform [72,73]	Adjunctive to anticoagulation, high cost, standard dose insufficient in severe cases	Cost; efficacy to be assessed for each patient with a functional assay before its use [82]	Not yet studied in HIT patients

Legend: aHIT, autoimmune HIT; IV, intravenous. For more details, please refer to the text and to the cited references.

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
