# Peer review of "Heparin-Induced Thrombocytopenia: A Review of New Concepts in Pathogenesis, Diagnosis, and Management"

_jcm, 2021, doi:10.3390/jcm10040683_

Round 1

Reviewer 1 Report

This is a review on “Heparin-induced thrombocytopenia (HIT): a review of new concepts in pathogenesis, diagnosis, and management” submitted for the for special JCM issue “The Latest Clinical Advances in Thrombocytopenia”. The authors, who are experts in this important area of basic (immunological) and clinical research, aim to summarize the new concepts in pathogenesis, diagnosis, and management of HIT. While multiple experimental and clinical investigation as well as major reviews have been published in the past this review submitted here succeeds to critically and thoroughly describe the most relevant aspects of HIT. This review is clearly of high interest for a broad readership. This referee has some minor comments to be considered.

  • The authors summarize that HIT is primarily caused by unfractionated heparin (UFH) and that there are alternatives to UFHs. Is there still any reason to use UFHs in the year of 2021?
  • The authors briefly mention HIT without heparin (spontaneous autoimmune HIT / aHIT) and refer to a recent review by Greinacher et al. It would be helpful for this review if regular HIT and aHIT could be briefly compared at the molecular level and resulting therapeutic approaches.    
  • The authors discuss that potent, short-acting and reversible platelet inhibitors (tirofiban, iloprost, cangrelor) have been initially examined to treat HIT for intraoperative “alternative” anticoagulation. Have these approaches also studied in other HIT situations? What about other platelet inhibitors?

Reviewer 2 Report

This review well describes the new concepts in pathogenesis, diagnosis and management of HIT. However, several sections should be improved as suggested below.

Specific comments

Pathophysiology section

2.a line 49-50:  Krauel et al (JTH 2008 DOI: 10.1111/j.1538-7836.2008.03171.x) reported an optimal PF4/H ratio of 20:1. Please discuss this finding.

2.a line 51:  “These mechanisms explain why fondaparinux is immunogenic without causing HIT… » This sentence should be tempered because fondaparinux can rarely induce true HIT syndromes.

2.b. line 56: A comma is missing between formed and ULCs

2.c and 2.d: The sections are confusing and poorly detailed. Thus, the immune response and antibody synthesis in HIT is not only mediated by complement activation and several other articles have been published on this topic. This section should be expanded or completely deleted.

2.d: The authors should develop and explain the mechanisms explaining the variability of platelet response to HIT antibodies especially the impact of FcgRIIA polymorphism.

2.e: page 3 line 85: The relative affinity of PF4 for various types of GAG could be discussed for a better understanding.

2.e: page 3 line 88: Is TF really secreted, as IL-8, or rather overexpressed? please clarify

2.e page 3: In the endothelial cells section, The new founding recently published by Johnston (Blood 2020; DOI: 10.1182/blood.2018881607) regarding the role of vWF on pathophysiology of HIT should be discussed. Thus, this study has been demonstrated that assembly of HIT immune complexes along VWF strings released by injured endothelium that might propagate the risk of thrombosis in HIT.  vWF could also be added to the figure 1.

In addition, on the figure 1, F(ab)2 receptors have been localized at the endothelial cells surface. What is the role of the F(ab)2 receptor in the pathophysiology of HIT, in particular on endothelial cell activation induced by anti-PF4/H antibodies ?

2.e page 3 line 100: 2.e page 3: “NETs formation » instead of « NETs activation »

2.g. : This section should better be placed above, regarding immune response to PF4.

Diagnosis section

3.a : Clinical pre-test probability

Il ne parle pas de l’évolution du cas CBP ou ecmo et de profile d’évaoltion des plaquettes. Cependant, dans les reco de l’ASH je ne vois pas cette distinction dans le score pré-clinic…

3.b :     The platelet activation tests section should be moved after the immunoassays in accordance with the diagnostic approach. For more details on platelet activation assays used for HIT diagnosis, you could refer the reader to a recent review  published in JCM (Tardy et al JCM 2020)

3.b.i line 182: “possibly containing anti-PF4/heparin antibodies”… the sentence is incorrect since platelet activation assays are performed after positive immunoassays. Thus, the word “activating or pathogenic” anti-PF4/H antibodies should be added.

3.b.i line 192-193: Please nuance this statement. These methods have not yet been validated by an appropriate study, but recent data clearly show that they could be of interest in raising sensitivity for the detection of platelet activating antibodies Nazy et al (JTH 2021; DOI: DOI: 10.1111/jth.15233) and Vayne et al (BJH 2017 DOI: 10.1111/bjh.14955). The concentration of PF4 used in these assays is another point of discussion, as large differences exist between the different methods, and can potentially impact the specificity of these tests.

3.3 page 9 line 259 : One of the limitations of these two approaches is that they have been validated with specific rapid IA, and are therefore not applicable to other rapid IAs. Please discuss.

  1. Management of acute HIT

Some passages are quite redundant with the table. You could probably lighten the table to make it fit on one page and make it more readable.

- Table 2: Potential side effects (such as allergic, metabolic or infectious) could be mentioned in cons of TPE

4.c: The part relative to 5B9 antibody should be reduced.

  1. Management of patients requiring cardiovascular surgery

However, in cardiac surgery patients, as recently outlined (Gruel Y, Anaesth Crit Care Pain Med. 2020; doi:10.1016/j.accpm.2020.03.01.), SRA may not be sensitive enough to formally eliminate the presence of potentially pathogenic antibodies in the patient’s plasma during cardiac surgery. This is all the more true since it has been shown that the addition of exogenous PF4 allows SRA to be positive, and cardiopulmonary bypass induces a significant and rapid increase in plasma PF4 concentration in operated patients.
